# Experimental Characterization and Multi-Factor Modelling to Achieve Desired Flow, Set and Strength of N-A-S-H Geopolymers

**DOI:** 10.3390/ma15165634

**Published:** 2022-08-16

**Authors:** Chaofan Yi, Yaman Boluk, Vivek Bindiganavile

**Affiliations:** Department of Civil and Environmental Engineering, University of Alberta, Edmonton, AB T6G 1H9, Canada

**Keywords:** compositional ratios, mutual interaction, geopolymerization, multi-factor models, mix design

## Abstract

The interaction between compositional ratios, namely, SiO_2_/Al_2_O_3_, Na_2_O/Al_2_O_3_, H_2_O/Na_2_O and the liquid-to-solid ratio, triggers mutual sacrifice between workability, setting time and strength for N-A-S-H geopolymers. The present study characterizes the mechanism underlying the effect of these compositional ratios and, in turn, develops guidelines for mixture design that requires a simultaneous and satisfactory delivery of these engineering properties. The experimental results show that an increase in either the SiO_2_/Al_2_O_3_, Na_2_O/Al_2_O_3_ or H_2_O/Na_2_O ratio raises the liquid-to-solid ratio, which in turn improves the workability of fresh mixtures. A continuous increase beyond 2.8 for the SiO_2_/Al_2_O_3_ ratio boosts its strength, but also significantly extends its final set. Lowering the Na_2_O/Al_2_O_3_ ratio from 1.3 to 0.75 raises the compressive strength significantly, while the shortest final set was seen at the median value, 1.0. A H_2_O/Na_2_O ratio of 9~10 yields the highest strength and the fastest final set simultaneously, due to the maximized degree of geopolymerization. Moreover, the accompanying sensitivity analysis indicates that the workability depends chiefly upon the H_2_O/Na_2_O ratio, the final setting time on the SiO_2_/Al_2_O_3_ ratio and, that the compressive strength relies on both of them. Also, this study proposes an optimal range of 2.8~3.6 for SiO_2_/Al_2_O_3_, 0.75~1.0 for Na_2_O/Al_2_O_3_ and 9~10 for H_2_O/Na_2_O to guarantee high strength, together with high flow and within the allowable final setting time. Furthermore, multi-factor predictive models are established with acceptable accuracy for practitioners to regulate oxide compositions in N-A-S-H geopolymers, which will guide future mixture design.

## 1. Introduction

It is well-established that the manufacturing of Portland cement results in inordinately high carbon emissions and is recognized as a significant contributor to global warming [1,2,3]. However, Portland cement is second only to potable water in per capita consumption with over 41 billion metric tons of concrete produced as of 2020 [4]. Hence, in recent decades, it has become an absolute necessity to reduce the use of Portland cement and natural aggregates, aiming to alleviate the associated carbon footprint [5,6]. In this regard, the usage of waste such as fly ash and lightweight aggregates has led to sustainable cement-based composites with comparable engineering properties [5,6]. More recently, alkali-activated systems have been identified as potential substitutes for Portland cement, due to the significantly low carbon emissions during manufacturing and their satisfactory mechanical performance. Specifically, a reduction in CO_2_ emissions up to 70~80% can be achieved [7,8]. Furthermore, while the strength is comparable with that of conventional Portland cement-based systems, alkali-activation often assures high early strength as well [9]. An alkali-activated system requires an aluminosilicate precursor and a strong alkali. The former can be any source of amorphous alumina and silica, often a combustion ash from an industrial or agricultural furnace, or a natural deposit. Prior studies have shown that metakaolin [10,11], slag [12,13], fly ash [14,15], red mud [16], zeolite [17] and sugarcane bagasse ash [18] are examples of eligible precursors. The Na_2_O-Al_2_O_3_-SiO_2_-H_2_O (N-A-S-H) geopolymer is a subset of the family of alkali-activated systems, which is usually produced with a precursor that is rich in silica and alumina but low in calcium content [5,7]. The alkaline environment required for its activation may be generated with either sodium hydroxide or potassium hydroxide in combination with their respective silicate in an aqueous solution [19]. The latter serves as a secondary source of silica to extend the oligomeric aluminosilicate chain [20]. The first step in the synthesis of alkali-activated systems is the dissolution of the solid aluminosilicate precursor in a strongly alkaline environment. This forms the tetrahedral AlO_4_ unit, which is thereafter linked to the tetrahedral SiO_4_ unit, through sharing an oxygen atom. In the meantime, the negatively charged [OH]^−^ ion supplied by the alkali-activating solution will attach to the oligomeric aluminosilicate chain, which extends the valence sphere. In this manner, the system achieves polycondensation [21]. On the other hand, although the tetrahedral AlO_4_ group is preferentially formed in the prevailing alkaline environment, it still carries a unit negative charge due to the donation of an electron from the shared oxygen atom. Thus, the alkali metal ions such as Na^+^, K^+^ and Ca^2+^ are essentially needed to balance this negative charge in order to attain chemical equilibrium [22,23]. Eventually, the oligomeric aluminosilicate framework that is generated will self-condense and form a more complex and stronger network, which may be expressed chemically as M_n_[-(Si-O_2_)_z_–Al–O]_n_· *w*H_2_O [24,25,26,27]. Here, *M* is the alkali metal ion sourced from the activator, and *n* represents the eventual degree of polycondensation. Additionally, *w* is the number of chemically bound water molecules, while *z* is the number of silicon atoms that constitute a single oligomeric aluminosilicate chain. The latter depends, in turn, upon the molar ratio of SiO_2_/Al_2_O_3_ contained in the raw materials. The choice of this ratio varies according to the desired setting time and strength [28]. As this value progressively increases from 1 to 3, a sialate, i.e., M_n_(-Si-O-Al-O-)_n_, a sialate-siloxo, i.e., M_n_(-Si-O-Al-O-Si-O-)_n_, or a sialate disiloxo, i.e., M_n_(-Si-O-Al-O-Si-O-Si-O)_n_, will be formed, respectively. (Note that the above atomic ratio translates into twice the corresponding oxide ratio, e.g., Si/Al = 1 is equivalent to SiO_2_/Al_2_O_3_ = 2).

It is widely reported that the engineering properties of N-A-S-H geopolymer systems are strongly dependent upon their chemical compositions and other mixing proportions. Among the compositional ratios, the role of SiO_2_/Al_2_O_3_ has been examined extensively. A higher SiO_2_/Al_2_O_3_ molar ratio usually promotes the polycondensation degree and, in turn, yields a higher compressive strength [29,30]. However, at the same time, the higher SiO_2_/Al_2_O_3_ molar ratio makes the self-condensation between silicon compounds more difficult than that occurring between silica and alumina and, therefore, the corresponding condensation rate is relatively slow [31]. Since the oligomeric aluminosilicate chain carries a single negative charge due to the tetrahedral AlO_4_ unit, the molarity of M^+^ should theoretically be equivalent to that of the aluminum species contained in the mixture in order to attain chemical equilibrium [32]. However, excessive M^+^ will hinder the geopolymerization [33]. Prior studies also indicate that both an excess and a deficit in the H_2_O/M_2_O molar ratio are detrimental to the properties of the evolving geopolymer. Whereas very low ratios adversely affect the stability of the aluminosilicate [33], an excessively high ratio hinders the structural integrity due to insufficient activation [34]. Besides the experimental investigations, some numerical studies have been conducted to predict the compressive strength of geopolymers. At present, the artificial neural network method is most frequently used to forecast mechanical properties [35,36,37,38]. For instance, Kamalloo et al. optimized SiO_2_/Al_2_O_3_, R_2_O/Al_2_O_3_ and H_2_O/R_2_O ratios as 3.6–3.8, 1.0–1.2, and 10–11, respectively, to achieve the maximum compressive strength [35], whereas the optimal combination was predicted as SiO_2_/Al_2_O_3_ = 2.90, Na_2_O/Al_2_O_3_ = 0.58, H_2_O/Na_2_O = 13.75 by Ghanbari et al. [37].

As summarized above, the influence of SiO_2_/Al_2_O_3_ molar ratio on the strength of geopolymer systems has been well-documented. However, the underlying mechanisms that detail the role of Na_2_O/Al_2_O_3_ and H_2_O/Na_2_O have not been comprehensively revealed. Additionally, their effects upon other equally vital properties such as the rheology and setting have not been systematically examined and, therefore, require further investigations. Notwithstanding the active research on N-A-S-H geopolymers, there exists very limited information to guide the mixture design to guarantee all of these attributes, namely, workability, setting and strength, simultaneously. It should be noted here that due to the mutual interactions between mixing design ratios, the best performance of one property, e.g., strength, is likely to be achieved by sacrificing other engineering properties. Furthermore, there is currently no set of accurate and explicit models available for practitioners to operate a flexible mix design for N-A-S-H geopolymers in accordance with varying engineering demands. Accordingly, the authors here proposed multi-factor models to predict workability, final set and compressive strength based on the compositional ratios, i.e., the liquid-to-solid, SiO_2_/Al_2_O_3_, Na_2_O/Al_2_O_3_ and H_2_O/Na_2_O ratios. In the meantime, a set of experimental characterizations was carried out, firstly, to form a sufficient dataset for model establishment and secondly, to clarify the underlying mechanism of these ratios affecting the individual engineering properties of geopolymers. Further, the sensitivities of workability, setting and strength to these compositional ratios are quantified by a variance-based analysis. When serving as a potential alternative to conventional Portland cement concrete used in civil engineering, the outcomes generated in this study are promising to guide the mixture design of N-A-S-H geopolymers for structural members with varying priorities upon workability, final set and strength. Together with the sensitivity analysis, the proposed multi-factor models shall serve as an efficient predictive tool for practitioners to conduct and validate their mix design for N-A-S-H geopolymers.

## 2. Experimental Program

### 2.1. Materials and Mix Proportions

A combination of sodium silicate and sodium hydroxide was used as the activator. The sodium silicate solution comprised approximately 40% sodium silicate compound and 60% deionized water, and the overall SiO_2_/Na_2_O modulus (molar ratio) was 3.2. The sodium hydroxide was sourced as solid pellets with a purity of 99%. The formation of N-A-S-H geopolymer requires high amorphous SiO_2_ and Al_2_O_3_ content [28]. A commercially sourced metakaolin was selected to serve as the aluminosilicate precursor. Its chemical composition, as determined by X-ray fluorescence (XRF), is listed in Table 1. As seen therein, this metakaolin is composed of 53.8% SiO_2_ and 43.8% Al_2_O_3_ by mass. Considering the molar mass of these two oxides, namely, 60 and 102, respectively, the corresponding SiO_2_/Al_2_O_3_ molar ratio may be calculated as approximately 2.1. The associated XRD and FTIR spectra for this precursor are presented in Figure 1. Notice the smooth hump in Figure 1a, centre at 2θ = 22.5°, alongside minor crystals identified as anatase and quartz. Three prominent peaks were detected in the FTIR spectrum at 1060 cm^−1^, 792 cm^−1^ and 434 cm^−1^; see Figure 1b. These are identified as the Si–O bonds in amorphous SiO_2_, tetrahedral AlO_4_ and T-O-T (T: Si or Al), respectively.

Three series of mixtures incorporating varying SiO_2_/Al_2_O_3_, Na_2_O/Al_2_O_3_ and H_2_O/Na_2_O molar ratios were designed, as listed in Table 2. In addition to measuring their flow, setting time and compressive strength, these mixtures were treated as representatives to be tested for temperature evolution, XRD, TGA, FTIR and SEM. An additional number of approximately 50 batches of specimens were made with different compositional ratios and examined for flow, setting time and compressive strength. Thus, a total of about 60 mixtures were prepared in all for this study.

### 2.2. Sample Preparation

The production of N-A-S-H geopolymer specimens starts with the preparation of the alkali-activating solution. It should be emphasized here that the solubility of sodium hydroxide pellets was about 111 g in 100 mL water under room temperature, here, ~20 °C. Therefore, for those mixtures made with a higher SiO_2_/Al_2_O_3_ molar ratio, such as 3.6 and 4.0, the water is predominantly sourced from the sodium silicate solution. Hence, a very small amount of extra water needs to be added manually. More importantly, the heat that evolves in the process of preparing the alkaline solution, especially for higher concentrations, poses potential risks of causticity. Thus, the sodium hydroxide pellets were blended with the sodium silicate solution along with extra distilled water to produce the alkali activator in solution. The beaker containing the blended activator solution was sealed and placed in a fume hood for 24 h, allowing the contents to cool down to room temperature. On the following day, the alkali activator in solution was first poured into the mixer and stirred for 60 s to ensure its homogeneity. Next, the solid precursor was added gradually to let it blend with the alkali-activating solution for 3~5 min until a homogeneous slurry was obtained. This mixture was then cast into plastic cylindrical moulds with dimensions of Φ50 mm × 100 mm height. The specimens, after demoulding were cured in air-tight plastic bags under ambient conditions to reach 28 days of maturity.

### 2.3. Test Protocols

The engineering properties were evaluated for various geopolymers. In this manner, the workability and setting time were tested as per ASTM C230/C230M-08 [39] and ASTM C191-08 [40], respectively. The compressive strength was measured for geopolymer specimens aged 28 days, as per ASTM C39/C39M-18 [41].

Further characterizations were also conducted in order to understand the mechanisms underlying the attendant effects of each oxide ratio on the above engineering properties. In this regard, the rheology of fresh mixtures was investigated using a Brookfield DV-II+ Programmable Viscometer that was fitted with a SC4-27 spindle. This viscometer operates within a specified range of viscosity between 0 to 1250 mPa·s with a constant rotation of 200 rpm. The temperature evolution during the setting process was investigated using thermography [28]. For each mixture, the fresh paste was first poured into a mould chamber (Φ50 mm × 100 mm) that could be closed with a tightly secured lid. The lid was bored with a hole in its centre to accommodate the thermometer with a functioning range of −50~300 °C. The mould was covered by a layer of polyurethane foam to minimize the temperature loss from the mixture. Once secure, the thermometer was inserted through the lid and the real-time temperature was recorded against time.

The mineral phases constituting the hardened composite were assessed through X-ray diffraction (XRD), of representative samples, using a copper Kα radiation beam (operated at 40 kV and 44 mA) with a step size of 2°/min, from 10° to 60° diffraction angle (2θ). A thermogravimetric analysis (TGA) was carried out using the Leco TGA 701 instrument that functions in a temperature range of 20~800 °C and at a heating rate of 10 °C/min under a nitrogen atmosphere. The chemical bonds involved in geopolymers were examined by using the iS50 Fourier transform infrared (FTIR) spectrometer coupled with a built-in attenuated total reflection (ATR) module. The microstructure of representative paste samples was examined under a field emission scanning electron microscope (FE-SEM) with a 15 kV accelerating voltage. The generated images were then binarized to show the voids and cracks as black and the solid geopolymer gel as fully white. The compactness of the geopolymer microstructure may, thus, be quantified by counting the black pixels.

## 3. Experimental Results and Analyses

### 3.1. Rheology of Fresh Geopolymer Mixtures

The rheological parameters of fresh geopolymer mixtures help explain the associated workability. We recall that the functional range of the rotational viscometer used here is 0~1250 mPa·s. However, all the mixtures listed in Table 2 exceeded this upper bound during the experimental trials. Hence, an independent batch of mixtures with suitably higher NaO_2_/Al_2_O_3_ and H_2_O/Na_2_O ratios was prepared, as shown in Table 3, and their rheological outcomes are presented in Figure 2. Notice that as the SiO_2_/Al_2_O_3_ molar ratio increases from 2.4 to 3.6, both the viscosity and the yield shear stress drop at first, reaching a minimum at 3.2; see Figure 2a,d. This is followed by a recovery for higher values of this ratio. Recall that the SiO_2_ content was raised through an increase in the Na_2_SiO_3_ and, as such, it increases the liquid content. Whereas the Na_2_SiO_3_ used here has a viscosity of 200~400 mPa·s at 20 °C, that of the NaOH and water is only 70 mPa.s and 1 mPa.s, respectively. Therefore, an increase in the viscosity and the yield shear stress may now be explained through the “competition” between the increased content of sodium silicate solution and the enlarged liquid-to-solid ratio. On the one hand, an increase in sodium silicate solution raises the overall viscosity of the alkali-activator solution since the viscosity of sodium silicate solution is significantly higher than NaOH and water. In turn, the viscosity of the fresh geopolymer slurry is also higher. On the other hand, the higher sodium silicate solution also leads to a greater liquid-to-solid ratio, which has long been recognized to make the slurry more flowable by alleviating the inter-particle friction [42]. The smaller mutual friction is supposed to reduce the viscosity of the slurry. Given the above, there may exist a competition between the above two effects. Based on these results, when increasing the SiO_2_/Al_2_O_3_ molar ratio from 2.4 to 3.2, the effect led by the increase in the liquid-to-solid ratio governs the rheology of the geopolymer mixture, as the content of sodium silicate is relatively small. However, with a further increase in SiO_2_/Al_2_O_3_ beyond 3.2, the sodium silicate content dominates over the presence of water alone; see Table 3. For the Na_2_O/Al_2_O_3_ and H_2_O/Na_2_O molar ratios, any change in their value is independent of the content of sodium silicate solution in the mixture. As a result, increasing either one of them simply raises the liquid-to-solid ratio. This, in turn, lowers both the viscosity and the yield shear stress of the fresh geopolymer mixture.

### 3.2. Temperature Evolution during Setting Process

N-A-S-H geopolymer mixtures show strong exothermicity and, therefore, the associated temperature changes are related to the reaction rate and degree [28,42]. Accordingly, the temperature–time history for each geopolymer mixture here is presented in Figure 3. To begin with, a significant difference, not only in the peak temperatures but also in its rate of rise, was witnessed for the series made with varying the SiO_2_/Al_2_O_3_ molar ratio. The mixture registering the quickest rate of increase, as well as the highest peak temperature (see Figure 3a), was that made with a SiO_2_/Al_2_O_3_ molar ratio of 2.8. However, both an increase and a decrease in this ratio led to progressively lower peaks and a slower rate of temperature rise. This indicates that the SiO_2_/Al_2_O_3_ molar ratio must be in an optimum range, otherwise, it will slow down the geopolymerization. It is likely due to the competition in the dissolution between the solid silica and alumina, in the precursor. At the lower end, the raw precursor is the sole source of aluminosilicate, and the progress of subsequent chemical reactions is dominated by the availability of dissolved silica and alumina. However, as the SiO_2_/Al_2_O_3_ molar ratio increases through the addition of sodium silicate, the dissolved alumina will quickly react with the liquid silicon groups from the activator to form the (-Si-O-Al-O-) chain [28]. On the other hand, when this molar ratio exceeds a certain value (here, 2.8) any further rise triggers a more complex polycondensation, i.e., the formation of (-Si-O-Al-O-Si-O-) [28]. This extends the reaction time and manifests in a slower temperature rise. Note that Davidovits reported a geopolymerization thermograph (in Chapter 8 within [28]) for N-A-S-H geopolymers made with different sources of metakaolin having varied chemical composition. Specifically, the one made with the larger SiO_2_-to-Al_2_O_3_ mass fraction also displays slower temperature evolution. However, this result is not comparable with the data generated in the present study, as the former was examined in an oven with the curing temperature set to 80 °C. For mixtures made with different Na_2_O/Al_2_O_3_ molar ratios, the generated peak temperatures differed slightly from one another; see Figure 3b. Of the values chosen in this study, the mixture made with a Na_2_O/Al_2_O_3_ = 1.0 showed a faster temperature evolution with a slightly higher peak than the rest. As mentioned earlier, each (-Si-O-Al-O-) or (-Si-O-Al-O-Si-O-) chain contains a single AlO_4_ unit and carries a negative charge. Therefore, an equivalent amount of Na^+^ is required in order to maintain the chemical equilibrium. This accounts for the slowing down of the geopolymerization process at lower Na_2_O/Al_2_O_3_ molar ratios. Additionally, at the higher range of this ratio, the excessive Na^+^ may deter polycondensation as per the reaction kinetics, since this reaction is also a NaOH regeneration process [28]; see Figure 4. Similarly, it also explains the slightly depressed geopolymerization in the case of a low H_2_O/Na_2_O ratio, manifesting in the reduction in both the rise and the peak temperature. On the other hand, at ratios higher than a threshold, there is too much water, which will dilute the alkali and lower the activation efficiency. Once again, this will reduce the exothermicity. Thus, as seen in Figure 3c, the rate of temperature rise and its peak are maximum for a H_2_O/Na_2_O ratio of 9.

### 3.3. X-ray Diffraction (XRD) and Thermogravimetric Analysis (TGA)

The XRD diffractograms and TG/DTG curves of the hardened geopolymer systems are shown in Figure 5a–c and d–f, respectively. One can see from Figure 5a that when SiO_2_/Al_2_O_3_ molar ratio is 2.1, numerous crystalline phases including faujasite, anatase, hydroxysodalite, Na-chabazite and quartz show up. This clearly indicates that for lower SiO_2_/Al_2_O_3_ molar ratios, geopolymerization is suppressed, forming Na-substituted zeolites instead [43]. Now, as this ratio rises through 2.8 and to 4, the peaks in the XRD trace that represent the crystalline phases reduce substantially. Simultaneously, a broad and clear diffuse hump appears, implying a greater degree of amorphicity. Connecting with the amorphous character of N-A-S-H networks, an increase in the SiO_2_/Al_2_O_3_ molar ratio favours this geopolymerization. Interestingly, the mixture displaying the better amorphicity also registers a sharper and taller differential thermogravimetry (DTG) peak positioned at 100~300 °C. These taller peaks are mainly connected to the dehydroxylation of the sodium aluminosilicate hydrate phase (N-A-S-H) [44]. As the SiO_2_/Al_2_O_3_ molar ratio drops through 2.8 and to 2.1, this peak becomes flatter and lower. It also shifts towards the higher end of the testing temperature. Whereas these Na-substituted zeolites lose their chemically bound water when raised from 100 °C to 300 °C, the associated DTG peak appears at a slightly higher temperature (~170 °C) than the (-Si-O-Al-O-Si-O-) chain that constitutes the N-A-S-H framework (~125 °C) [28]. Clearly, this is attributed to other crystalline “impurities”, as noted in the XRD spectra in Figure 5a–c, namely, faujasite, anatase, Na-chabazite and hydroxysodalite.

For the Na_2_O/Al_2_O_3_ molar ratio, the two mixtures at the lower end of the Na_2_O/Al_2_O_3_ molar ratio registered fewer crystalline phases and displayed the tell-tale hump of amorphous constituents, as evident from Figure 5b. On the other hand, the mixture containing a Na_2_O/Al_2_O_3_ ratio of 1.3 displays significant crystalline phases, an outcome of the formation of zeolite at high alkali content [45]. This agrees well with the TGA findings presented in Figure 5e. As seen therein, the mixture exhibiting the superior amorphicity, once again, registers the sharper DTG peak assigned to the dehydroxylation of the N-A-S-H network. Furthermore, the corresponding DTG peak shifts towards the lower end of the temperature scale as the Na_2_O/Al_2_O_3_ ratio decreases down to 0.75.

Figure 5c maps the XRD trace for mixtures made with varying the H_2_O/Na_2_O ratio. All three traces show only limited crystalline phases. Nevertheless, a small yet significant difference is observed by locating the centre of the diffuse hump in each trace. When the H_2_O/Na_2_O ratio moves from either extreme of 9 and 12 to the median 10, the value of 2θ corresponding to the diffuse peak increases from 28.43° and 28.49°, respectively, to a local maximum of 29°. According to Yuan et al. [46], such a shift is indicative of an increase in the amorphicity of the geopolymerized system despite a limited presence of crystals. In the case of the H_2_O/Na_2_O ratio, it is evident that this has no bearing upon the nature of the reaction products formed in N-A-S-H geopolymers. The corresponding TGA data shown in Figure 5f illustrate that, regardless of the H_2_O/Na_2_O ratio, identical DTG plots were obtained, especially at 100–300 °C, associated with the dehydroxylation of the N-A-S-H framework.

### 3.4. Fourier Transform Infrared Spectroscopy (FTIR)

In their mature state, the geopolymerized samples were examined through FTIR, as presented in Figure 6. As seen therein, a small band located around 560~570 cm^−1^ is ascribed to the external linkage vibrations of the TO_4_ in the double rings of zeolite [47,48]. In addition, it has been widely reported that the most significant signal, assigned to the asymmetric stretching vibrations of the geopolymer band Si-O-T, is usually found between 850~1250 cm^−1^ [49,50]. Here, T is the 4-coordinated atom, Si or Al. More importantly, the degree of geopolymerization is associated with the shift of this main band [50]. Further, the band positioned at ~1645 cm^−^^1^ is attributed to the bending vibrations H-O-H, implying the presence of structural water in the reaction products [48]. Now, as seen from Figure 6a, the mixture made with a SiO_2_/Al_2_O_3_ ratio of 4.0 registered the highest wavenumber associated with the Si-O-T band; in turn, the weakest signal was assigned to the TO_4_ in the double rings of zeolite. This was followed by those for SiO_2_/Al_2_O_3_ of 2.8 and 2.1, respectively. From the above, it is clear that an increase in the SiO_2_/Al_2_O_3_ ratio promotes geopolymerization and simultaneously depresses the formation of zeolites. This agrees well with the XRD traces in Figure 5a. A continuous increase in the Na_2_O/Al_2_O_3_ molar ratio was seen to deter the associated geopolymerization, as evident from the principal Si-O-T bands shifting towards the smaller wavelength in the FTIR spectra; see Figure 6b. As expected, the wavelength ascribed to the zeolite formation increases as this ratio rises from 0.75 to 1.3. Assuming a perfectly activated system, the optimum Na_2_O/Al_2_O_3_ ratio is theoretically calculated as 1.0 in order to achieve the charge balance. However, in practice, the raw aluminosilicate precursor may not be completely activated. The XRD traces and the detected TO_4_ band assigned to zeolites in the FTIR spectrum also lead to the same inference. Thus, it is not surprising that the actual optimum Na_2_O/Al_2_O_3_ is, in fact, a bit lower than the predicted value of 1. As opposed to the variation in the S-O-T band seen with varying the SiO_2_/Al_2_O_3_ or Na_2_O/Al_2_O_3_ ratio, the FTIR spectra for the series prepared with varying the H_2_O/Na_2_O molar ratio do not register a significant change; see Figure 6c. Nevertheless, the mixture made with a H_2_O/Na_2_O ratio of 10 is seen to register the highest wavenumber for the Si-O-T band among the three values of this molar ratio. Once again, when the H_2_O/Na_2_O molar ratio is excessive, it deters geopolymerization. When it is too low, the excessive alkalinity in the mixture depresses polycondensation, as explained through Figure 4.

In order to interpret the structural changes referred to above, the FTIR spectra were deconvoluted for the range of 800~1250 cm^−^^1^ that corresponds to the Si-O-T band, using the PeakFit 4.12 (SeaSolve Software Inc., San Jose, CA, USA) software with Gaussian peak shapes [48]. The associated fitting procedure was performed in accordance with the related literature [48,51]. The following peaks are considered: Peak I (850~860 cm^−1^) to Si-OH bending; Peak II (930 cm^−1^) to Si-O-T in Q^2^; Peak III (950~960 cm^−1^) to asymmetric stretching vibrations of nonbridging oxygen (NBO) sites; Peak IV (990 ~1000 cm^−1^) to Si-O-T in a 3-dimentional N-A-S-H network; Peak V (1080~1090 cm^−1^) to Si-O-Si of silica gels; and Peak VI (1150~1160 cm^−1^) to Si-O-T of unreacted metakaolin. The FTIR deconvolution peaks for each mixture are now presented in Figure 7, with a regression coefficient *R*^2^ beyond 0.997. In addition, the relative peak areas are quantified in Figure 8. It is evident that an increase in the SiO_2_/Al_2_O_3_ ratio from 2.1 to 4.0 leads to a reduced area for Peak III but an increase in areas for both Peaks IV and V. Clearly, this corresponds to a greater degree of geopolymerization [51]. Further, a slight increase is also seen in the area of Peak VI for SiO_2_/Al_2_O_3_ = 4.0, i.e., in the unreacted metakaolin particles. This implies a slight increase in the unreacted metakaolin for an increase in the sodium silicate content. This is not surprising, since adding more sodium silicate solution will lower the relative amount of sodium hydroxide in the mixture and, thus, depress the dissolution of the precursor due to the slightly reduced alkalinity [52]. Nevertheless, the rise in unreacted metakaolin is minor when compared to the rise in the areas of Peaks IV and V. In sum, therefore, a rise in the SiO_2_/Al_2_O_3_ ratio favours a higher degree of geopolymerization.

On the other hand, a continuous increase in Na_2_O/Al_2_O_3_ leads to a drop in areas of both Peaks IV and V. Further, a noticeable rise is witnessed for Peak II when this ratio increases to 1.3. This corresponds to an increase in the Q^2^ silicate. Together, they imply a depressed degree of geopolymerization. Finally, in the case of the H_2_O/Na_2_O ratio, the highest area for Peaks IV and V are noticed for the mixture with H_2_O/Na_2_O = 10. As noted in Figure 5c,f, it confirms that there exists an optimum alkali concentration to yield the maximum degree of geopolymerization.

### 3.5. Scanning Electron Micrographs

The microstructure of representative mixtures from the three series was examined under SEM. The images captured at 300X magnification are shown in Figure 9. Those mixtures prepared with varying the SiO_2_/Al_2_O_3_ ratio are illustrated in Figure 9a–c. From a ratio of 2.1 through 4.0, while microcracks are visible, it is wider in the former, progressively becoming narrow in the latter. The higher the SiO_2_/Al_2_O_3_ ratio, the more compact and the more homogenous the microstructure. This illustrates that the condensed structure, resulting from the higher degree of geopolymerization, has the greater structural integrity. Figure 9d–f represents mixtures made with varying the Na_2_O/Al_2_O_3_ molar ratio. While microcracks appear in the sample made with the Na_2_O/Al_2_O_3_ of 0.75, the corresponding crack widths are significantly smaller than in the other two mixtures containing the higher Na_2_O/Al_2_O_3_ ratios. Particularly, the mixture containing a Na_2_O/Al_2_O_3_ of 1.3 displays the widest microcrack in this series. This may be attributed to the following: (i) the continuous increase in this ratio depresses the geopolymerization degree, as confirmed in previous XRD and FTIR outcomes. Note here that the lower degree of geopolymerization usually indicates the lower structural integrity of formed N-A-S-H networks; (ii) the larger Na_2_O/Al_2_O_3_ corresponds to a greater liquid-to-solid ratio (see Table 1), and the increased liquid content may degrade the texture of hardened geopolymer networks. Given these, it may not be surprising to note the widest microcracks appear in the case of Na_2_O/Al_2_O_3_ = 1.3. Figure 9g–i shows the variation in the H_2_O/Na_2_O molar ratio. Clearly, the mixture made with the H_2_O/Na_2_O = 10 registers the best microstructure, as evident from the smallest microcracks. Both a decrease and an increase in this ratio appear to degrade the microstructure of N-A-S-H geopolymers, due to the depressed degree of geopolymerization. The former may be led by an excessive alkali concentration, as the geopolymerization is a regeneration process of Na^+^ and OH^−^ (see Figure 4), while the latter is simply due to the inefficient alkali activation. Furthermore, the quantification for the microcracks and voids was conducted after binarizing these images. According to Figure 9j–l, the lowest area fraction of cracks and voids is found for SiO_2_/Al_2_O_3_ = 4.0, Na_2_O/Al_2_O_3_ = 0.75 and H_2_O/Na_2_O = 10, respectively, across each series.

### 3.6. Workability, Final Setting Time and Compressive Strength

The workability is expressed here in the form of flow diameter. As seen in Figure 10a, the workability increases substantially with an increase in any of the three compositional ratios. While it is clear that an increase in the H_2_O/Na_2_O ratio directly implies a higher water content, an increase in the SiO_2_/Al_2_O_3_ or Na_2_O/Al_2_O_3_ ratios also, indirectly, implies a higher liquid-to-solid fraction. This is because the SiO_2_ and Na_2_O may only be varied by varying the activator, which in this case is introduced as a solution. It is illustrated in Figure 10b, wherein an increase in any of these three oxide ratios leads to an increase in the liquid-to-solid ratio. For a fixed amount of dry powder in the mixture, a higher liquid-to-solid ratio will act as a lubricant [52] and, thus, lead to a greater flowability. Recall that for the rheological parameters presented in Figure 2, an increase in the Na_2_O/Al_2_O_3_ or H_2_O/Na_2_O ratio corresponds to a decrease in the viscosity and yield shear stress of the fresh mixture. This agrees well with the superior flow seen in Figure 10a. As explained earlier, it is due to the Na_2_SiO_3_ being substantially more viscous, in comparison with NaOH or water, at 20 °C. Beyond a critical SiO_2_/Al_2_O_3_ ratio, found here as 3.2, the geopolymer slurry turns more viscous due to a higher sodium silicate content, despite an associated increase in the liquid-to-solid ratio. However, for an increase in the SiO_2_/Al_2_O_3_ ratio, the flow is governed by the liquid-to-solid ratio, and manifests as a continuous improvement in flow diameter.

The effect of the compositional ratios in the mixture components upon the setting time is shown in Figure 10c. It is seen that at a low SiO_2_/Al_2_O_3_ ratio, i.e., 2.1 in the present study, the setting process is delayed because all the silica originates from the metakaolin, and so it takes time to allow the aluminosilicate to dissolve and then be activated. On the other hand, an excess of SiO_2_/Al_2_O_3_ ratio, as noticed beyond 2.8 in this series, implies a transformation of the product from the relatively simple sialate (-Si-O-Al-O-) structure to the more complex sialate-siloxo (-Si-O-Al-O-Si-O-) framework. The latter is a more interconnected network that requires some time to form and so, once again, extends the setting time so that there exists an optimum range for the SiO_2_/Al_2_O_3_ ratio to ensure the minimum setting time. In Figure 3, the mixture made with the optimum SiO_2_/Al_2_O_3_ ratio may be taken as that which displays the highest peak temperature and the quickest temperature evolution. The H_2_O/Na_2_O ratio directly reflects the alkalinity of the system. So, under very high alkalinity, the excessive Na^+^ and OH^−^ will deter the polycondensation (see Figure 4), manifesting as a delayed final set. In contrast, the excessive H_2_O/Na_2_O ratio dilutes the alkali concentration and, thus, reduces the activation efficiency. An optimum range for the H_2_O/Na_2_O ratio is found around 8–10 to ensure the shortest duration for the final set. As for the initial setting, an increase in this ratio extends the initial setting time, possibly due to the higher liquid-to-solid ratio. In contrast to the two oxide ratios discussed above, the effect seen for the Na_2_O/Al_2_O_3_ ratio upon the final setting time was relatively minor. Nevertheless, a value of about 1.0 makes both initial and final set occur somewhat sooner. A deficit in the Na_2_O/Al_2_O_3_ ratio leads to insufficient Na^+^ required to balance the negative charge carried by ${\mathrm{AlO}}_{4}^{-}$. On the other hand, earlier characterizations reported that an excess in this ratio is likely to depress the N-A-S-H formation and even cause the transformation into various crystals. This, thereafter, corresponds to a delay in the set of mixture. Note that all these setting outcomes agree well with the temperature evolution curves presented in Figure 3.

As seen from Figure 10d, within the examined range of 2.1~4.0, an increase in the SiO_2_/Al_2_O_3_ ratio boosts the associated strength. Recall that the mixture made with a lower SiO_2_/Al_2_O_3_ ratio witnessed a substantial presence of crystalline phases, namely, various Na-substituted zeolites, in the XRD trace. The same mixture also registered fewer peak areas ascribed to the Si-O-T band in 3D N-A-S-H network under FTIR inspection. These together confirm that increasing the SiO_2_/Al_2_O_3_ ratio restricts the formation of other crystalline impurities (principally the Na-substituted zeolites) and also improves the compactness of the N-A-S-H framework. This eventually manifests as higher mechanical strength. An increase in the Na_2_O/Al_2_O_3_ ratio implies a lower compressive strength, with the associated optimum range lying between 0.75~1.0, according to Figure 10d. Note that a continuous increase in Na_2_O will result in the formations of other zeolite crystals, which strongly depresses the polycondensation of N-A-S-H geopolymers, as evident from the associated XRD, TGA and FTIR outcomes. Furthermore, the alumina contained in the raw precursor may not completely be associated with the geopolymer. Thus, the optimum value for the Na_2_O/Al_2_O_3_ molar ratio was somewhat lower than the theoretical value, viz. 1.0. In the case of the H_2_O/Na_2_O ratio, where a lower value denotes a stronger alkalinity, excessive alkali concentration may also suppress regeneration of NaOH and, in turn, hinder the geopolymerization progress, according to Figure 4. On the other hand, when this ratio is too high, it deters activation efficiency and restricts geopolymerization. As such, the compressive strength was highest for a H_2_O/Na_2_O ratio of 10; see Figure 10d. It is worth noting here that unlike the SiO_2_/Al_2_O_3_ and Na_2_O/Al_2_O_3_ ratios, varying the H_2_O/Na_2_O ratio has minimal impact upon the products resulting in the N-A-S-H geopolymer.

### 3.7. Sensitivity Analysis

An effective variance-based sensitivity analysis [53] was conducted in this study to quantify the influence of these oxide ratios on workability, final set and compressive strength. It is illustrated mathematically by Equations (1) and (2). The sensitivity index, *S_i_*, for each factor is defined as the proportion of the effective partial variance, *V_ei_*, in the total effective variance, *V_te_*. An increment coefficient, *α_ij_*, is introduced to eliminate the disturbance and normalize the traditional variance. The oxide ratio, *x*, and the engineering property, *y*, are the input and output variables, respectively. In this study, the engineering property denotes either the flow, the setting time or the compressive strength. *n* denotes the size of the dataset, with *m* being the median value of each factor. The data used for this analysis are listed in Table 4, and the resulting sensitivity indices are illustrated in Figure 11. As seen therein, the workability of the fresh mixture is most sensitive to the H_2_O/Na_2_O molar ratio, while the compressive strength of N-A-S-H geopolymers is more dependent upon, firstly, the H_2_O/Na_2_O and secondly, the SiO_2_/Al_2_O_3_ molar ratios. Further, the SiO_2_/Al_2_O_3_ molar ratio dominates the setting process, as evident from its sensitivity index beyond 95%.
(1)$$V_{e}{}_{i}=\frac{1}{n-1}\sum_{j=1}^{n}\frac{{\left(y_{ij}-y_{im}\right)}^{2}}{\alpha_{ij}}\;\mathrm{with}\;\alpha_{ij}=\frac{x_{ij}-x_{im}}{x_{im}}$$
(2)$$S_{i}=\frac{V_{ei}}{V_{te}}=\frac{V_{ei}}{\sum_{i=1}^{l}V_{ei}}\in \;[0,1]$$

## 4. Establishment of Multi-Factor Models

### 4.1. Find the Relationship between Variables and Predicted Outcomes

The multi-factor models require sufficient experimental observations first. Accordingly, together with 12 mixtures shown in Table 2, about 60 mixtures were produced. Of these, 50 mixtures (~85%), as shown in Table 5, were selected randomly to establish the proposed multi-factor models. The remaining 10 datasets (~15%), as illustrated in Table 6, were used later to validate the proposed models. The process to establish the multi-factor model is as follows: First, a correlation between each *explanatory* variable (*x_i_*) and the output (*y*) is established. This study aims to investigate the effect of compositional ratios on various engineering properties of N-A-S-H-type geopolymers. Thus, the three oxide ratios, namely, SiO_2_/Al_2_O_3_, Na_2_O/Al_2_O_3_ and H_2_O/Na_2_O, are selected as three *explanatory* variables. Note that a change in any one of the above ratios will automatically alter the liquid-to-solid ratio. Hence, it is taken here as the fourth *explanatory* variable when developing the corresponding multi-factor models for workability, final set and compressive strength. According to the experimental results (Figure 10), the flow diameter of freshly produced N-A-S-H geopolymer increases with an increase in the amount of any of the three compositional ratios, and an approximate linear correlation may be assumed. Hence, a preliminary model for flow diameter is described as Equation (3). However, this model does not account for the coupling between the selected variables. Thus, the mutual combinations between three oxide ratios are now taken into account. Consequently, the multi-factor model for the flow diameter of N-A-S-H-type geopolymers is updated as Equation (4). With regard to the final set, the actual experimental observations shown in Table 5 confirmed that there indeed exist optimum values within the range of examined compositional ratios to yield the quickest setting process. They are approximately 2.4~2.8 for SiO_2_/Al_2_O_3_, 0.75~0.8 for Na_2_O/Al_2_O_3_ and 9~10 for H_2_O/Na_2_O. Hence, a parabolic function was adopted in this study to simulate the correlation between oxide ratios and the setting time. However, as with flow, the setting time was once again modelled using a linear equation to capture the effect of the liquid-to-solid ratio. The uncoupled multi-factor model for the final set is shown in Equation (5). This first-generation model was updated further after accounting for the mutual interactions between the four *explanatory* variables, and is expressed in Equation (6). An increase in the SiO_2_/Al_2_O_3_ ratio or a drop in the Na_2_O/Al_2_O_3_ ratio within the examined range yields a higher strength; see Figure 10. Again, due to their monotonic behaviour, a linear correlation is assumed for their influence on the compressive strength. The optimum H_2_O/Na_2_O is found to be around 9~10 for N-A-S-H geopolymers. Thus, the correlation between this ratio and the compressive strength may be approximated through a parabolic function. The multi-factor models for compressive strength are shown uncoupled in Equation (7) and in Equation (8) with coupling considered.
(3)$$F_{flow}=a_{1}x_{1}+a_{2}x_{2}+a_{3}x_{3}+a_{4}x_{4}+a_{5}$$
(4)$$\begin{array}{c} F_{flow}=f_{1}+f_{2}\;with\\ f_{1}=a_{1}x_{1}+a_{2}x_{2}+a_{3}x_{3}+a_{4}x_{4}+a_{5}\\ f_{2}=a_{6}x_{4}(x_{1}+a_{7}x_{2}+a_{8}x_{3})+a_{9}x_{1}(x_{2}+a_{10}x_{3}+a_{11}x_{2}x_{3}) \end{array}$$
(5)$$F_{set}=a_{1}{\left(x_{1}-2.4\right)}^{2}+a_{2}{\left(x_{2}-0.8\right)}^{2}+a_{3}{\left(x_{3}-9\right)}^{2}+a_{4}x_{4}+a_{5}$$
(6)$$\begin{array}{c} F_{set}=f_{1}+f_{2}\;with\\ f_{1}=a_{1}{\left(x_{1}-2.4\right)}^{2}+a_{2}{\left(x_{2}-0.8\right)}^{2}+a_{3}{\left(x_{3}-9\right)}^{2}+a_{4}x_{4}+a_{5}\\ f_{2}=a_{6}(x_{1}-2.4)(x_{2}-0.8)(x_{3}-9)(x_{4}+a_{7}) \end{array}$$
(7)$$F_{strength}=a_{1}x_{1}+a_{2}x_{2}+a_{3}{\left(x_{3}-9\right)}^{2}+a_{4}x_{4}+a_{5}$$
(8)$$\begin{array}{c} F_{strength}=f_{1}+f_{2}\;with\\ f_{1}=a_{1}x_{1}+a_{2}x_{2}+a_{3}{\left(x_{3}-9\right)}^{2}+a_{4}x_{4}+a_{5}\\ f_{2}=a_{6}(x_{1}+a_{7})(x_{2}+a_{8})(x_{4}+a_{9})+a_{10}x_{1}x_{2}x_{4}{\left(x_{3}-9\right)}^{2} \end{array}$$

Once again, *x_i_* denotes the individual *explanatory* variable: SiO_2_/Al_2_O_3_ (Si/Al), Na_2_O/Al_2_O_3_ (Na/Al), H_2_O/Na_2_O (H/Na), and the liquid-to-solid ratio when *i* increases from 1 to 4, respectively.

### 4.2. Determine Regression Coefficients for Multi-Factor Models

Each unknown coefficient is linearly expanded with a couple of regression coefficients, *b_i_*. The size of the coefficient vector, *b*, depends on the number of items after variable separation, *z*. The expanded model may be stated as a matrix to include all *n* sets of data, as given in Equation (9). The residual error, *ε*, between the predicted and the actual values is computed by a *n* × 1 vector; see Equation (10). Based on the least square method, the optimum solution to Equation (9) will be found when the sum of the squares of the residuals, Equation (11), is a minimum. In this manner, the minimum is determined by setting the first-order derivative of *S*(*ε*) equal to zero; see Equation (12). Note that the *S*(*ε*) has scalar values, whereas *ε* is a column vector incorporating *n* components. So, we will have *n* first-order derivatives and follow the convention to arrange them in a column vector. Eventually, the optimum solution to the vector of coefficients is obtained, as shown in Equation (13). A proof that *ε* minimizes the sum of the squares of the residuals is given by Equation (14). As seen therein, the second-order derivative of *S*(*ε*), i.e., the Hessian matrix, is determined as a positive definite matrix.
(9)$$Y=\left(\begin{array}{c} Y_{1}\\ Y_{2}\\ \vdots \\ Y_{n} \end{array}\right)=\left(\begin{array}{cccc} X_{11} & \cdots & X_{1z} & 1\\ X_{21} & \cdots & X_{2z} & 1\\ \vdots & \vdots & \vdots & \vdots \\ X_{n1} & \cdots & X_{nz} & 1 \end{array}\right)\left(\begin{array}{c} b_{1}\\ b_{2}\\ \vdots \\ b_{n} \end{array}\right)$$
(10)ε=y−Xb
(11)$$S(\varepsilon )=\sum \varepsilon_{i}=y^{\prime }y-y^{\prime }Xb-b^{\prime }X^{\prime }y+b^{\prime }X^{\prime }Xb$$
(12)$$\frac{\partial S(\varepsilon )}{\partial b}=-2X^{\prime }y+2X^{\prime }Xb=0$$
(13)$$b={\left(X^{\prime }X\right)}^{-1}X^{\prime }y$$
(14)$$\frac{\partial^{2}S(\varepsilon )}{\partial^{2}b}=2X^{\prime }X$$

The optimum estimate of coefficient vectors for each proposed multi-factor model is solved using a subset that incorporates 85% of the experimental dataset. The proposed multi-factor models are now given below: Equations (15) and (16) for the flow diameter, Equations (17) and (18) for the final set, and Equations (19) and (20) for the compressive strength. Figure 12 compares the outcome from the proposed multi-factor models and the actual experimental observations. One can see that regardless of whether these models are coupled or not, the points distribute uniformly around the linearly fitted line. The error is quantified by evaluating the corresponding coefficient of determination, *R*^2^, according to Equations (21) and (22).
(15)$$F_{flow}=54.5\frac{\mathrm{Si}}{\mathrm{Al}}+183.4\frac{\mathrm{Na}}{\mathrm{Al}}+12.9\frac{H}{\mathrm{Na}}+57.1\frac{L}{S}-305.4$$
(16)$$\begin{array}{ll} F_{flow} & =145.4838\frac{\mathrm{Si}}{\mathrm{Al}}-353.055\frac{\mathrm{Na}}{\mathrm{Al}}+17.4084\frac{H}{\mathrm{Na}}+1148.87\frac{L}{S}-807.22\\ & -275.98\frac{L}{S}(\frac{\mathrm{Si}}{\mathrm{Al}}-0.783\frac{\mathrm{Na}}{\mathrm{Al}}+0.1924\frac{H}{\mathrm{Na}})+7.3366\frac{\mathrm{Si}}{\mathrm{Al}}(\frac{\mathrm{Na}}{\mathrm{Al}}+2.2237\frac{H}{\mathrm{Na}}+1.2745\frac{\mathrm{Na}}{\mathrm{Al}}\frac{H}{\mathrm{Na}}) \end{array}$$
(17)$$F_{set}=194.39{\left(\frac{\mathrm{Si}}{\mathrm{Al}}-2.4\right)}^{2}+266.32{\left(\frac{\mathrm{Na}}{\mathrm{Al}}-0.8\right)}^{2}+4.96{\left(\frac{H}{\mathrm{Na}}-9\right)}^{2}-58.77\frac{L}{S}+188.31$$
(18)$$\begin{array}{ll} F_{set}= & 275.915{\left(\frac{\mathrm{Si}}{\mathrm{Al}}-2.4\right)}^{2}+592.326{\left(\frac{\mathrm{Na}}{\mathrm{Al}}-0.8\right)}^{2}+5.3867{\left(\frac{H}{\mathrm{Na}}-9\right)}^{2}-20.936\frac{L}{S}+129.88\\ & -24.598(\frac{\mathrm{Si}}{\mathrm{Al}}-2.4)(\frac{\mathrm{Na}}{\mathrm{Al}}-0.8)(\frac{H}{\mathrm{Na}}-9)(\frac{L}{S}+6.389) \end{array}$$
(19)$$F_{strength}=39.3\frac{\mathrm{Si}}{\mathrm{Al}}+58.3\frac{\mathrm{Na}}{\mathrm{Al}}-0.2{\left(\frac{H}{\mathrm{Na}}-9\right)}^{2}-75.06\frac{L}{S}-37.8$$
(20)$$\begin{array}{ll} F_{strength}= & 101.9\frac{\mathrm{Si}}{\mathrm{Al}}+24.3647\frac{\mathrm{Na}}{\mathrm{Al}}-1.2137{\left(\frac{H}{\mathrm{Na}}-9\right)}^{2}-53.849\frac{L}{S}-177.3826\\ & +44.8786(\frac{\mathrm{Si}}{\mathrm{Al}}-2.4)(\frac{\mathrm{Na}}{\mathrm{Al}}-2.17)(\frac{L}{S}-0.1565)+0.3367(\frac{\mathrm{Si}}{\mathrm{Al}})(\frac{\mathrm{Na}}{\mathrm{Al}})(\frac{L}{S}){\left(\frac{H}{\mathrm{Na}}-9\right)}^{2} \end{array}$$
(21)$$R^{2}=1-\frac{\mathrm{SSR}}{\mathrm{SST}}=\frac{b^{\prime }X^{\prime }NXb}{y^{\prime }Ny}$$
(22)$$N=I-\frac{1}{n}ll^{\prime }$$

Here, *SST* and *SSR* denote the total variation and the residual variation in *y*, respectively; *I* is the identity matrix, l is the unit vector, and $l^{\prime }$ represents the inverse vector.

### 4.3. Model Justification

The three uncoupled models register lower *R*^2^ values: 0.79 for flow diameter, 0.87 for the final set and 0.83 for compressive strength. Upon coupling, the corresponding indices improved to 0.92, 0.95 and 0.88, respectively. Note further that another subset containing 15% of the “hitherto unseen” data was employed to examine the independence of the proposed multi-factor models upon the training process. The proposed coupled models were validated against this dataset in Figure 13. It is seen that the predicted data set falls within ±20% of this experimental set. As summarized above, the proposed multi-factor predictive models, even without coupling, promise guidelines for future N-A-S-H geopolymer mixture design.

## 5. Further Discussion

The present study shows that the interaction between these oxides triggers mutual sacrifice between workability, setting time and strength of N-A-S-H geopolymers. For example, although a continuous increase in the SiO_2_/Al_2_O_3_ molar ratio boosts the compressive strength of N-A-S-H geopolymers, it extends the final set considerably once this ratio exceeds a threshold. Furthermore, an increase in H_2_O/Na_2_O molar ratio beyond an upper bound appears to damage the structural integrity and strength while it contributes to the workability. As per ASTM C150/C150M-19 [54], the final setting time recommended for Portland cement should not exceed 375 min. Additionally, going by the workability criterion for geopolymers [55], a fresh geopolymer mixture with a median flow diameter larger than 180 mm may be viewed as highly workable. The experimental observations in this study show that those mixtures produced with a liquid-to-solid ratio beyond 1.1 could easily achieve this diameter. Further, extreme oxide ratios may hinder the strength development even though the corresponding mixture registered satisfactory workability and setting time, e.g., Na_2_O/Al_2_O_3_ = 1.3 (SiO_2_/Al_2_O_3_ = 2.8 and H_2_O/Na_2_O = 11). Taking this together, the authors recognize that the mix design for N-A-S-H geopolymers must simultaneously guarantee the desired workability, setting and strength. As seen in both Figure 10 and Table 5, when the SiO_2_/Al_2_O_3_ ratio falls within 2.8~3.6, it results in satisfactory compressive strength alongside acceptable final setting time and workability. When the actual Na_2_O/Al_2_O_3_ ratio falls slightly below the theoretical value of unity, it achieves a quicker final set and a higher compressive strength. The optimum range for this ratio was found to lie in 0.75~1.0 in the present study. A H_2_O/Na_2_O ratio between 9~10 guarantees a high flowability for the N-A-S-H mixture without undermining the other two engineering properties. Together, the above range of oxide ratios offers an optimal design for N-A-S-H geopolymers to achieve the most practical combination of workability, final set and compressive strength. If a specific application puts a premium on only one of three properties, the sensitivity results now provide an insight into the design strategy. Adjusting the H_2_O/Na_2_O ratio is the most effective means to improving workability, while controlling the SiO_2_/Al_2_O_3_ ratio imparts the greatest efficiency towards shorter setting time. The mechanical strength may be adjusted by regulating the above two ratios simultaneously. Thus, the proposed multi-factor models are effective checks for required fresh and hardened properties prior to the actual design of N-A-S-H geopolymer mixtures.

## 6. Conclusions

This study investigated the mechanisms that underlie the effects of compositional oxide ratios on the workability, setting time and strength of N-A-S-H geopolymers. The experimental observations demonstrate that, when suitably designed, the N-A-S-H geopolymer displays excellent fresh and hardened properties, and is able to serve as an eligible alternative to conventional Portland cement concrete used for casting structural members. Further, multi-factor models are proposed to predict the above fresh and hardened properties for N-A-S-H geopolymers as affected by varying the oxide compositions. Based on the results, the following specific conclusions may be drawn:(1).An increase in the Na_2_O/Al_2_O_3_ or H_2_O/Na_2_O ratio reduces the viscosity of the fresh geopolymer mixture due to a corresponding increase in the liquid-to-solid ratio. There was an accompanying drop in the yield shear stress. Together, this leads to a greater flowability. However, there exists an optimum SiO_2_/Al_2_O_3_ ratio to obtain the lowest viscosity. Whereas, at lower values, the system is dominated by sodium silicate, once this ratio exceeds 3.2, the liquid-to-solid ratio dominates and eventually raises the flowability of the system;(2).In N-A-S-H geopolymers, the setting time is associated with the exothermicity during the geopolymerization, which was found most sensitive to the SiO_2_/Al_2_O_3_ ratio. The mixture made with a SiO_2_/Al_2_O_3_ ratio of about 2.8 registered the highest peak temperature and the fastest temperature rise, both of which coincide with its fastest setting time;(3).A deficient SiO_2_/Al_2_O_3_ ratio and an excessive Na_2_O/Al_2_O_3_ ratio will, in either case, deter geopolymerization. Additionally, they promote significant zeolite formation. This was evident from the reduced amorphicity in XRD, a wider but much lower DTG peak assigned to depolymerization of N-A-S-H in TGA, and also a smaller peak area ascribed to the Si-O-T band in the N-A-S-H framework in FTIR. The H_2_O/Na_2_O ratio at its optimum value, found to be 9~10 here, led to the highest amorphicity and degree of polycondensation;(4).The proposed multi-factor models capture the effect of each oxide component in the mixture design upon the possible flow, final setting time and compressive strength. When the mutual interactions between the different oxide ratios are considered, the coefficient of determination for flow improved from 0.79 to 0.92, for setting time, from 0.87 to 0.95 and from 0.83 to 0.88 for the compressive strength. More importantly, these proposed models may serve as a predictive tool to conduct and validate mixture design for N-A-S-H geopolymers with varying priorities upon workability, final set and compressive strength;(5).A satisfactory combination of compositional ratios to simultaneously achieve the desired workability, final set and compressive strength may fall within SiO_2_/Al_2_O_3_ = 2.8~3.6, Na_2_O/Al_2_O_3_ = 0.75~1.0 and H_2_O/Na_2_O = 9~10. The workability of the fresh N-A-S-H mixture is most sensitive to the H_2_O/Na_2_O ratio, while the setting time is predominantly governed by the SiO_2_/Al_2_O_3_ ratio. The compressive strength is sensitive to the H_2_O/Na_2_O ratio and, to a lesser extent, the SiO_2_/Al_2_O_3_ ratio as well.

## Figures and Tables

**Figure 1 materials-15-05634-f001:**
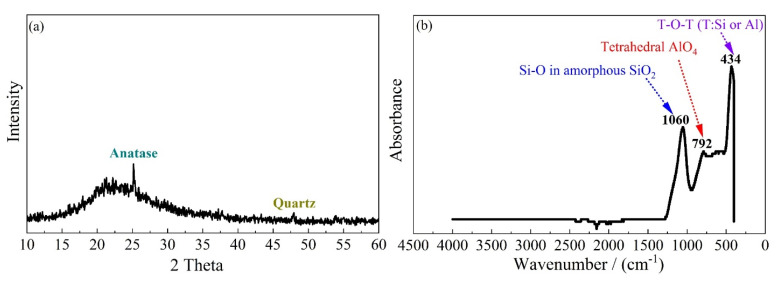
(**a**) XRD and (**b**) FTIR spectra of the metakaolin precursor.

**Figure 2 materials-15-05634-f002:**
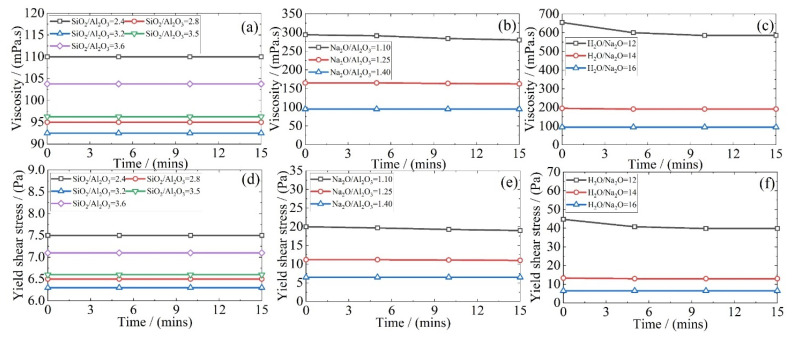
Rheological properties of fresh geopolymer mixtures made with varying compositional ratios: (**a**–**c**) viscosity and (**d**–**f**) yield shear stress.

**Figure 3 materials-15-05634-f003:**
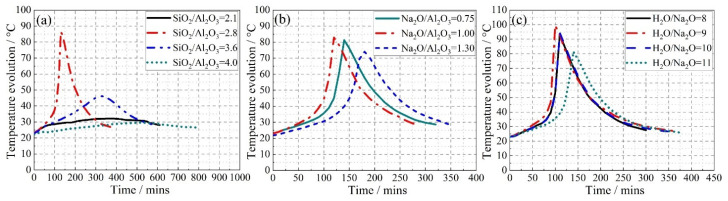
Temperature evolution as a function of time for fresh geopolymer mixture made with varying (**a**) SiO_2_/Al_2_O_3_, (**b**) Na_2_O/Al_2_O_3_, (**c**) H_2_O/Na_2_O molar ratios.

**Figure 4 materials-15-05634-f004:**
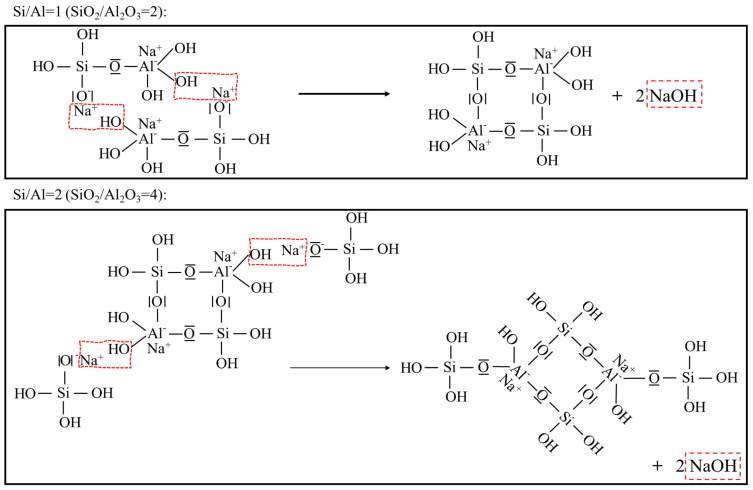
NaOH regeneration during the polycondensation for N-A-S-H geopolymer.

**Figure 5 materials-15-05634-f005:**
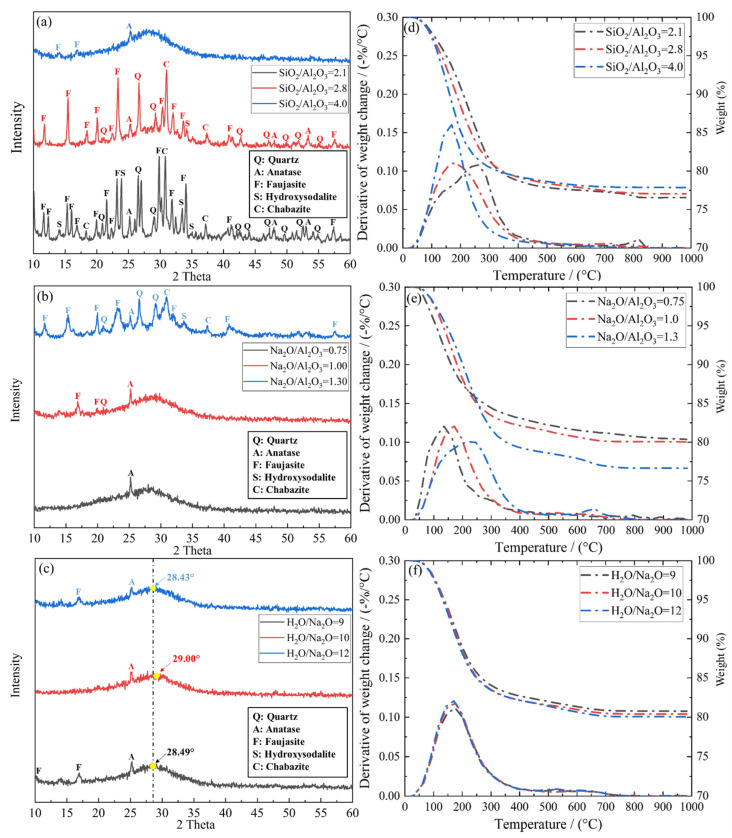
(**a**–**c**) XRD and (**d**–**f**) TGA outcomes of hardened geopolymers made with varying SiO_2_/Al_2_O_3_, Na_2_O/Al_2_O_3_ and H_2_O/Na_2_O molar ratios.

**Figure 6 materials-15-05634-f006:**
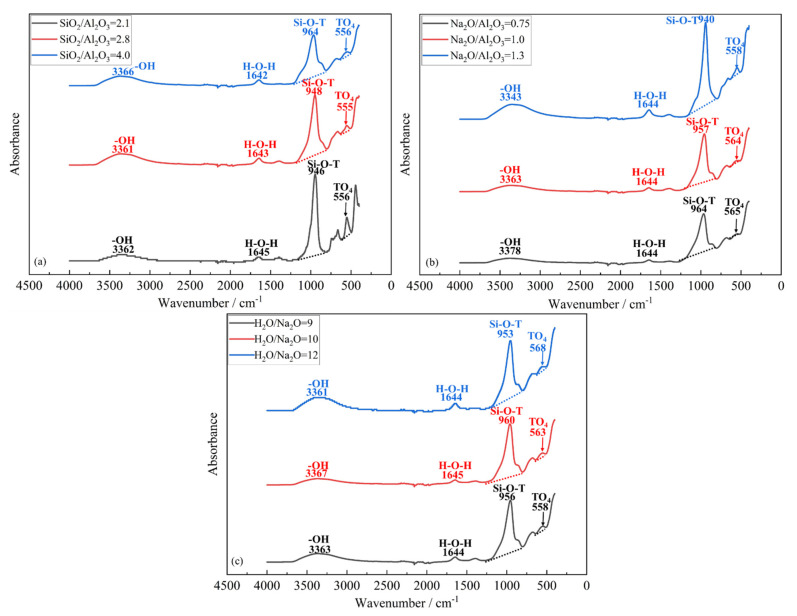
FTIR spectra of hardened geopolymers made with varying (**a**) SiO_2_/Al_2_O_3_, (**b**) Na_2_O/Al_2_O_3_ and (**c**) H_2_O/Na_2_O molar ratios.

**Figure 7 materials-15-05634-f007:**
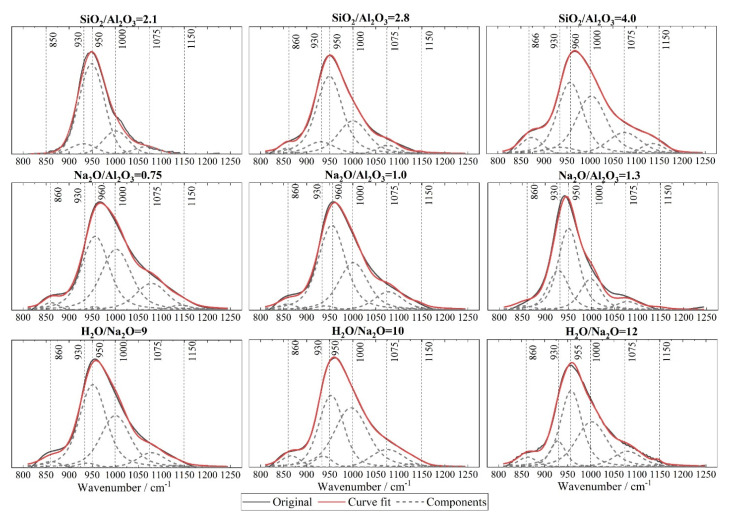
FTIR spectral deconvolutions of the main Si-O-T stretching band positioned at 800~1250 cm^−1^ for geopolymers reported in Figure 6.

**Figure 8 materials-15-05634-f008:**
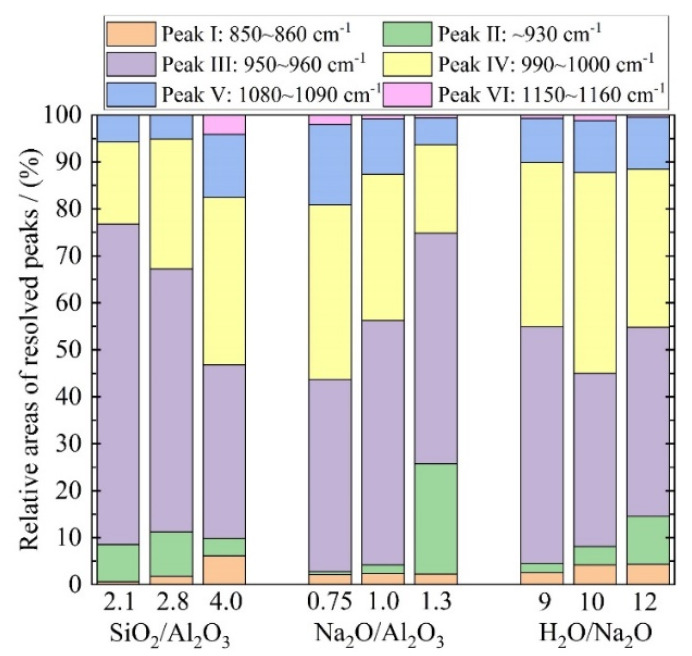
Relative areas of the deconvoluted component peaks within the main Si-O-T band.

**Figure 9 materials-15-05634-f009:**
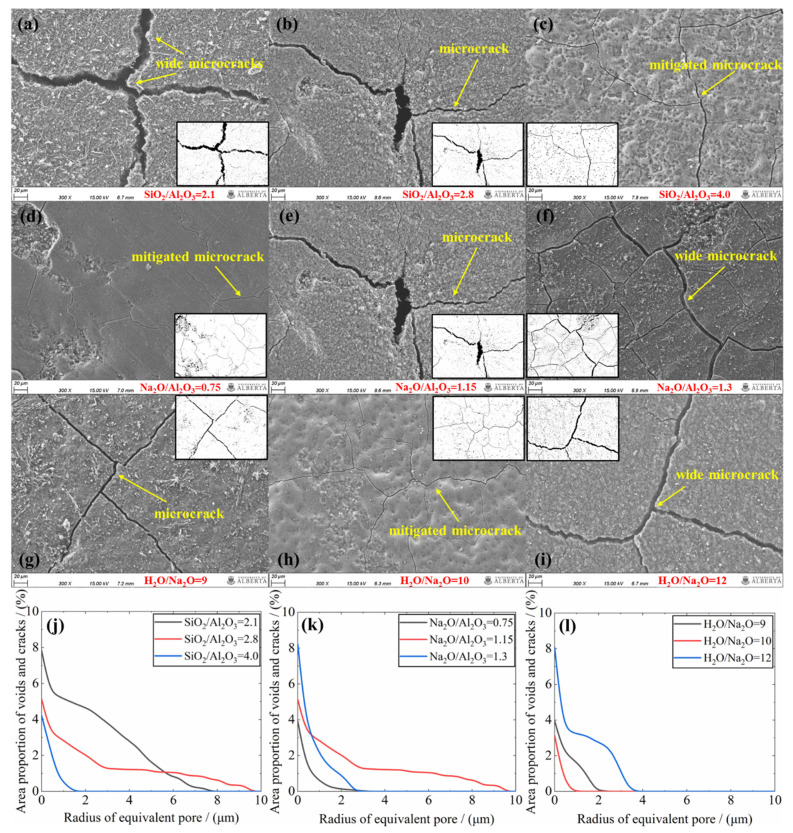
SEM images of geopolymer mixture made with varying (**a**–**c**) SiO_2_/Al_2_O_3_, (**d**–**f**) Na_2_O/Al_2_O_3_ (**g**–**i**) H_2_O/Na_2_O and (**j**–**l**), their quantified voids and cracks.

**Figure 10 materials-15-05634-f010:**
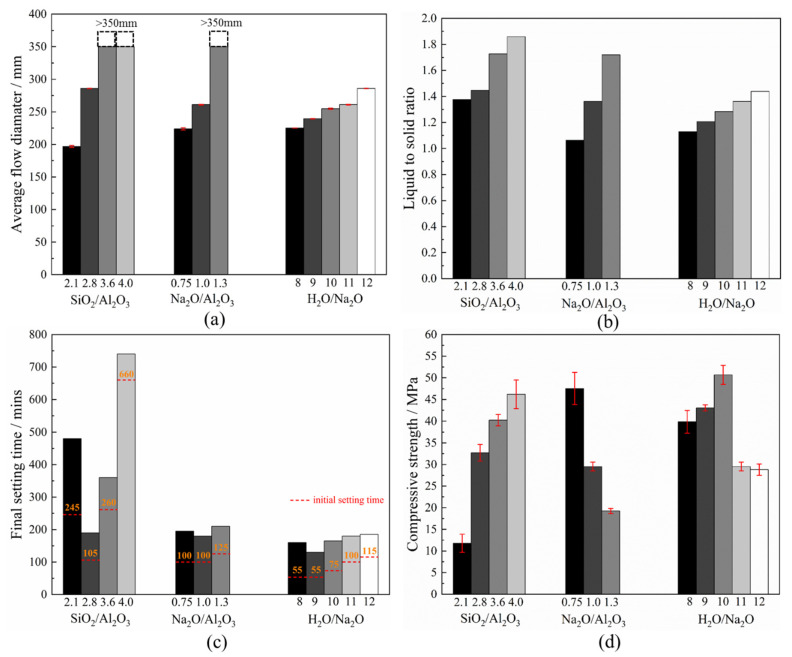
(**a**) Workability, (**b**) liquid-to-solid ratio, (**c**) setting time and (**d**) compressive strength of geopolymers made with varying compositional ratios (>350 mm indicates that the diameter of mixture is more than 350 mm after 25 blows on flow table).

**Figure 11 materials-15-05634-f011:**
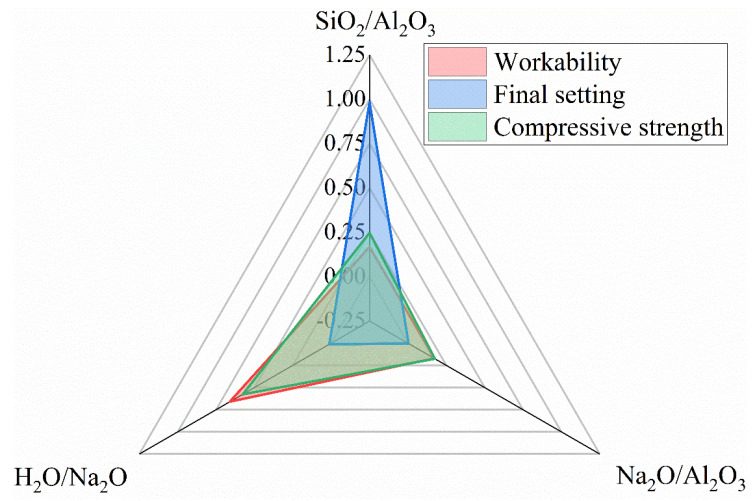
Sensitivities of workability, final set and compressive strength to compositional ratios.

**Figure 12 materials-15-05634-f012:**
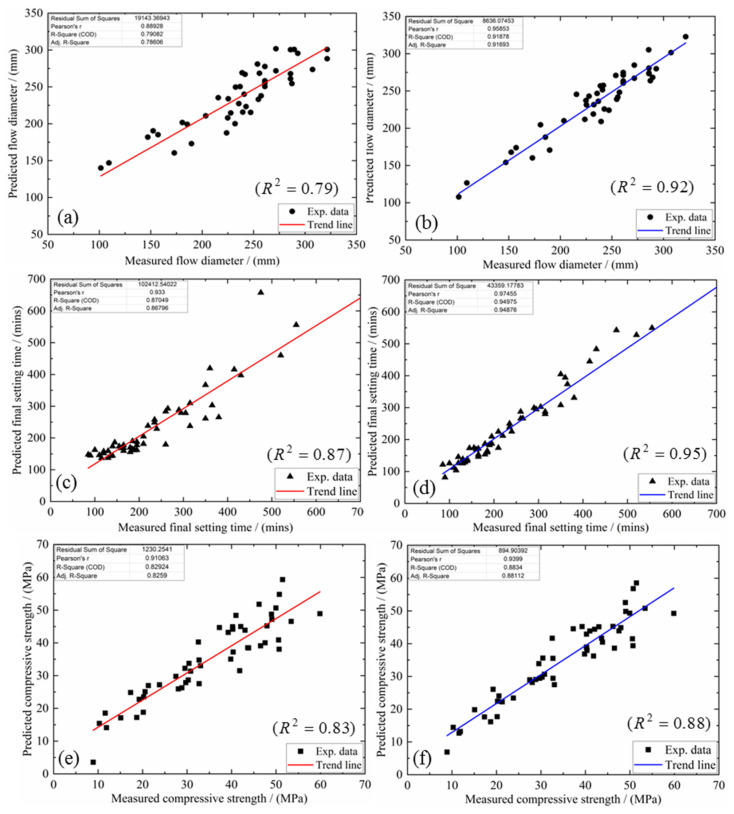
Comparing predicted results with actual observations for (**a**,**b**) flow diameter, (**c**,**d**) final set and (**e**,**f**) compressive strength.

**Figure 13 materials-15-05634-f013:**
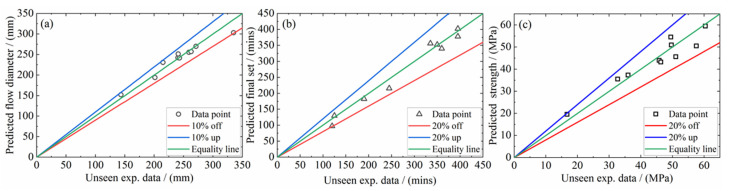
Comparing predicted results with validation dataset for (**a**) flow diameter, (**b**) final setting time and (**c**) compressive strength.

**Table 1 materials-15-05634-t001:** Chemical composition of metakaolin.

SiO_2_	Al_2_O_3_	TiO_2_	Fe_2_O_3_	P_2_O_5_	Na_2_O	K_2_O	CaO
53.8%	43.8%	0.9%	0.5%	0.4%	0.3%	0.2%	0.1%

**Table 2 materials-15-05634-t002:** Mix proportions of N-A-S-H geopolymers for engineering properties and further characterizations.

SiO_2_/Al_2_O_3_ (Molar Ratio)	Na_2_O/Al_2_O_3_ (Molar Ratio)	H_2_O/Na_2_O (Molar Ratio)	Metakaolin (g)	Sodium Silicate (g)	NaOH Pellet (g)	Water (g)	Liquid-to-Solid
2.1	1.15	11	500	6.63	196.86	484.96	1.377
2.8	1.15	11	500	320.98	160.76	400.76	1.540
3.6	1.15	11	500	680.23	119.50	63.96	1.727
4.0	1.15	11	500	837.41	104.89	0	1.885
2.8	0.75	11	500	320.98	92.02	118.37	1.063
2.8	1.00	11	500	320.98	134.98	224.70	1.361
2.8	1.30	11	500	320.98	186.54	352.29	1.720
2.8	1.00	8	500	320.98	134.98	108.71	1.129
2.8	1.00	9	500	320.98	134.98	147.37	1.207
2.8	1.00	10	500	320.98	134.98	186.04	1.284
2.8	1.00	11	500	320.98	134.98	224.70	1.361
2.8	1.00	12	500	320.98	134.98	263.37	1.439

Note: Liquid components include sodium silicate and dissolved NaOH in water; Solids represent metakaolin.

**Table 3 materials-15-05634-t003:** Proportioning the N-A-S-H paste mixtures for rheology test.

SiO_2_/Al_2_O_3_ (Molar Ratio)	Na_2_O/Al_2_O_3_ (Molar Ratio)	H_2_O/Na_2_O (Molar Ratio)	Metakaolin (g)	Sodium Silicate (g)	NaOH Pellet (g)	Water (g)	Liquid-to-Solid
2.4	1.40	16	50	14.14	22.43	77.77	2.287
2.8	1.40	16	50	32.10	20.37	66.55	2.380
3.2	1.40	16	50	50.06	18.31	55.32	2.474
3.5	1.40	16	50	63.53	16.76	46.90	2.544
3.6	1.40	16	50	68.02	16.25	44.09	2.556
2.8	1.10	16	50	32.10	15.22	47.99	1.906
2.8	1.25	16	50	32.10	17.79	57.27	2.143
2.8	1.40	16	50	32.10	20.37	66.55	2.380
2.8	1.40	16	50	32.10	20.37	44.90	1.947
2.8	1.40	16	50	32.10	20.37	55.72	2.164
2.8	1.40	16	50	32.10	20.37	66.55	2.380

**Table 4 materials-15-05634-t004:** Representative outcomes used for sensitivity analysis.

SiO_2_/Al_2_O_3_ (Molar Ratio)	Na_2_O/Al_2_O_3_ (Molar Ratio)	H_2_O/Na_2_O (Molar Ratio)	Flow Diameter (mm)	Final Setting Time (min)	Compressive Strength (MPa)
2.1	0.95	11	152.3	-	-
2.4 *			203.5	-	-
2.8			257.3	-	-
3.1			261.0	-	-
2.1	1.15	11	-	480	11.78
2.8 *			-	190	32.71
3.6			-	360	40.23
4.0			-	740	46.19
2.1	0.75	11	101.5	-	-
	0.95 *		152.3	-	-
	1.15		196.9	-	-
	1.25		247.7	-	-
2.8	0.75	11	-	195	47.54
	1.00 *		-	180	29.53
	1.15		-	190	32.71
	1.30		-	210	19.24
2.8	1.00	8	225.1	160	39.83
		9 *	239.5	130	43.03
		10	254.7	165	50.68
		11	261.0	180	29.53

Note: * denotes the median value.

**Table 5 materials-15-05634-t005:** Training dataset for establishing multi-factor models and predicted outcomes.

SiO_2_/Al_2_O_3_	Na_2_O/Al_2_O_3_	H_2_O/Na_2_O	L/S	Tested Flow (mm)	Predicted Flow (mm)	Tested Set (min)	Predicted Set (min)	Tested Strength (MPa)	Predicted Strength (MPa)
2.1	0.75	11	0.899	101.5 ± 1.2	107.8	140	144	17.3 ± 2.9	17.6
2.1	0.95	11	1.138	152.3 ± 1.7	167.8	185	172	11.6 ± 0.9	12.7
2.2	0.75	11	0.923	109.3 ± 0.7	126.6	100	131	21.3 ± 2.8	22.2
2.3	1.00	9	1.090	147.1 ± 1.0	154.1	115	124	33.0 ± 0.8	27.5
2.3	1.00	10	1.167	185.5 ± 0.8	188.0	130	131	23.8 ± 3.3	23.4
2.3	1.00	12	1.322	225.4 ± 0.2	231.3	160	179	10.3 ± 1.2	14.5
2.3	1.00	14	1.476	255.7 ± 0.5	241.7	220	269	8.9 ± 0.5	6.9
2.4	0.75	11	0.969	172.9 ± 0.2	160.1	120	123	30.8 ± 2.2	30.7
2.4	0.95	11	1.208	203.5 ± 0.4	210.0	135	129	20.6 ± 0.9	24.0
2.4	1.15	11	1.447	285.9 ± 0.5	280.6	180	184	20.2 ± 1.7	17.7
2.4	0.80	11	1.029	189.6 ± 1.8	170.7	85	120	27.5 ± 2.9	29.0
2.4	1.00	11	1.268	242.7 ± 0.2	225.7	140	139	20.3 ± 2.0	22.4
2.4	1.20	11	1.507	307.3 ± 0.1	301.5	195	205	18.7 ± 1.8	16.2
2.6	0.88	11.5	1.210	247.2 ± 1.8	224.2	120	136	30.3 ± 3.2	29.8
2.7	0.90	9	1.086	157.1 ± 1.9	173.9	90	128	39.2 ± 1.0	45.2
2.7	0.90	10	1.149	180.7 ± 1.0	204.6	110	126	43.8 ± 2.7	40.5
2.7	0.90	12	1.288	215.7 ± 1.0	245.5	165	155	29.7 ± 2.6	29.4
2.7	0.90	14	1.427	238.9 ± 0.7	256.9	235	227	15.1 ± 1.3	19.8
2.8	0.95	11	1.302	257.3 ± 0.7	248.2	165	149	30.4 ± 0.2	35.6
2.8	1.15	11	1.540	285.7 ± 0.2	305.2	190	171	32.7 ± 1.9	29.4
2.8	0.75	11	1.063	223.8 ± 1.6	211.9	195	172	47.5 ± 3.7	43.9
2.8	0.80	11	1.122	232.0 ± 0.8	219.0	155	162	43.6 ± 3.3	41.7
2.8	1.00	11	1.361	261.0 ± 0.5	260.6	180	150	29.5 ± 1.0	33.9
2.8	1.20	11	1.600	321.5 ± 0.8	322.7	195	184	28.0 ± 2.0	28.2
2.8	1.00	8	1.129	225.1 ± 0.2	170.8	160	182	39.8 ± 2.6	46.0
2.8	1.00	9	1.207	239.5 ± 0.3	209.0	130	162	43.0 ± 0.7	45.2
2.8	1.00	10	1.284	254.7 ± 0.9	238.9	165	151	50.7 ± 2.2	39.4
2.8	1.00	12	1.439	285.9 ± 0.2	274.0	185	160	28.8 ± 1.3	29.1
2.9	0.99	11	1.368	287.3 ± 0.1	263.7	145	167	39.8 ± 0.6	36.8
3.1	0.75	11	1.133	224.9 ± 0.6	237.1	315	267	53.4 ± 2.8	50.8
3.1	0.95	11	1.372	261.0 ± 0.7	263.3	240	221	32.5 ± 1.1	41.7
3.2	0.75	11	1.156	227.8 ± 1.2	243.0	380	310	49.0 ± 0.9	52.6
3.2	0.80	11	1.216	235.7 ± 0.9	246.7	350	293	49.1 ± 0.6	49.8
3.2	0.80	12.7	1.321	261.0 ± 1.7	271.2	365	343	46.5 ± 2.1	38.7
3.2	1.00	11	1.455	261.0 ± 1.1	274.7	235	249	50.6 ± 0.9	41.5
3.3	1.02	10.5	1.466	253.8 ± 0.7	270.8	260	296	48.0 ± 2.3	44.9
3.3	0.95	11	1.418	271.7 ± 0.2	267.1	290	297	37.3 ± 2.4	44.6
3.3	1.00	9	1.323	232.5 ± 2.1	231.6	305	339	50.8 ± 2.6	56.8
3.3	1.00	10	1.401	241.7 ± 1.7	257.4	295	309	59.9 ± 1.6	49.3
3.5	1.00	11	1.482	293.0 ± 0.4	279.8	350	383	50.0 ± 0.8	49.3
3.6	0.85	13	1.501	285.9 ± 0.5	273.3	520	527	40.2 ± 3.2	39.0
3.6	0.85	10	1.303	236.9 ± 0.7	236.2	430	485	51.5 ± 0.5	58.6
3.6	0.95	11.5	1.525	289.5 ± 0.5	268.2	415	445	41.0 ± 1.9	43.9
2.4	1.30	11	1.626	/	/	260	255	11.9 ± 0.3	13.2
2.8	1.30	11	1.720	/	/	210	218	19.2 ± 0.6	26.1
3.2	1.30	11	1.813	/	/	265	267	41.8 ± 1.7	36.2
3.6	1.15	11	1.727	/	/	360	407	40.2 ± 1.3	42.9
3.8	0.95	13	1.682	/	/	555	558	40.3 ± 3.5	38.7
4.0	1.20	12	1.973	/	/	475	533	42.1 ± 0.4	44.4
4.0	1.15	11	1.885	/	/	740	621	46.2 ± 3.3	45.3

**Table 6 materials-15-05634-t006:** Predicted outcomes for mixtures in the validating dataset.

SiO_2_/Al_2_O_3_	Na_2_O/Al_2_O_3_	H_2_O/Na_2_O	L/S	Tested Flow (mm)	Predicted Flow (mm)	Tested Set (min)	Predicted Set (min)	Tested Strength (MPa)	Predicted Strength (MPa)
2.2	0.85	11	1.042	143.53 ± 2.5	152.3	125	136	16.7 ± 1.3	19.6
2.6	0.90	10	1.126	201.40 ± 0.4	194.2	120	115	36.0 ± 3.2	37.4
2.7	1.00	11	1.275	240.87 ± 0.7	251.5	190	141	32.7 ± 2.4	35.5
3.0	0.85	11	1.229	240.54 ± 0.5	242.2	245	205	45.8 ± 3.3	44.0
3.4	0.90	10	1.312	243.13 ± 0.8	241.7	350	361	49.5 ± 1.7	54.6
3.4	0.80	10	1.201	214.90 ± 0.8	230.7	395	376	60.4 ± 4.3	59.5
3.4	1.00	10	1.424	263.03 ± 0.4	257.3	360	357	57.5 ± 2.7	50.5
3.4	1.20	10	1.648	335.14 ± 0.2	303.1	335	382	51.1 ± 2.1	45.6
3.4	0.85	12	1.388	271.13 ± 0.5	270.0	395	388	46.3 ± 1.2	43.2
3.6	0.85	11	1.369	259.02 ± 1.1	255.5	450	489	49.7 ± 1.5	51.0

## Data Availability

All data derived from this study are presented in the enclosed figures and tables.

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
