# Peer review of "Experimental Characterization and Multi-Factor Modelling to Achieve Desired Flow, Set and Strength of N-A-S-H Geopolymers"

_materials, 2022, doi:10.3390/ma15165634_

Round 1
Reviewer 1 Report
This paper investigated the influences of compositional ratios on workability, setting time and strength of Na2O-Al2O3-SiO2-H2O (N-A-S-H) geopolymers. The topic is of potential interest for readers. However, some of the results need further discussion and justification. In addition, some specific issues to address include:
(1) According to Figure 3(a), the temperature evolution of the fresh geopolymer mixture was extremely sensitive to the SiO2/Al2O3 value. Have the authors found similar results in the references?
(2) Page 13, the SEM images in Figure 9 were not well taken, and it is not convincible to discuss the microcracks and microstructure based on those images.
(3) Regarding the setting time, have the authors tested the initial setting time? The setting time of geopolymer is usually shorter than Portland cement, which might restrict its application. Why did the author preferred “quick set” in the abstract, Line 23?
(4) Please go through the manuscript carefully to revise some minor mistakes, for example, Line 75, “A higher SiO2/Al2O3 molar ratio promotes usually promotes…”; and Line 148, “extra water need be added…”
Reviewer 2 Report
COMMENTS
There are some comments that authors must address before this manuscript can be considered for publishing in materials.
- line 122-123. Please explain how to get the molar ratio of SiO2/Al2O3 of 2.1 from the table 1 !
- Figure 1 (a) for XRD, why the peaks of anatase and quartz in metakaolin precursor was defined only one and small peak, it‘s usually have several crystalline peaks of Anatase and quartz.
- Line 200. Please explain the used of “antagonism” in scientific words or give scientific reason the increase in the viscosity and the yield shear stress associated with sodium silicate solution and the liquid-to- solid ratio.
- Line 378-380. Please give reason why at a molar ratio Na2O/Al2O3 of 1.3, the microcracks is wider and pores.
- It appears obviously in SEM result that the microstructures of geopolymer mixture made with H2O/Na2O of 9 is the biggest compared with others and seems agglomerated. Why its happened.
- If there is Energy Dispersive X-Ray (EDX), it is better to add to the manuscript, in order to clarify the content of the elements and impurities in the geopolymer mixture.
Reviewer 3 Report
The paper deals with the developing of the guidelines for the mix-design of N-A-S-H geopolymer products to meet the satisfactory required engineering properties including desired flow, set and strength. The paper is globally well written and the experiments of design, the predicted results are interesting. However, the minor points below should be revised to improve the paper quality before publication.
The authors will read the entire the manuscript and remove some errors typing and grammatical mistakes.
The introduction and possibly conclusion should be enriched with information on the potential practical applications of the prepared materials.
The reviewer's suggestion: Minor revision
Reviewer 4 Report
Dear Authors,
Thank you for the opportunity to review this article.
My overall impression of the article is good, the topic is needed and it looks like a well done research.
Here is an evaluation of each section:
The title is too long and even if the authors want to express the whole idea, they should be able to be concise.
The abstract is written in a very specific way, but that is its disadvantage - it is completely uninteresting and indigestible. The abstract needs to be rewritten to make the benefit clearer even to less familiar readers.
The introduction is well written on the topic, but I would recommend increasing the continuity for a general construction audience. Thus, I would recommend expanding the citations to include, for example:
doi.org/10.1088/1755-1315/444/1/012021
doi.org/10.3390/math10020229
The prerequisites for the experimental program are deified clearly. I commend the extensive chemical descriptions.
It is great that you have prepared 66 mixtures. That is impressive.
The descriptions of the experiments are clear. Sample preparation is also very clear.
There is nothing surprising about the results - this is a practical demonstration where the hypothesis is clearly stated and the results are clear. The authors have demonstrated the importance of mixtures and the best ratios. The discussion is ok.
The model has a relatively low determinant - yes 0.79 to 0.95 are not small numbers, but I would expect a higher bestfit trend line. But the authors discuss and defend the accuracy well.
The conclusions are according to the results.
First page above the title: articel
The article contains a number of typos - please check it vigorously.
All the formulas have a strange format.
Figure 11 is too small.
The formulas on pages 20 and 21 are not in a good format - it looks odd.
Round 2
Reviewer 1 Report
The comments have been properly responded and the manuscript can be published in present form.